# Strategies of Modelling Incident Outcomes Using Cox Regression to Estimate the Population Attributable Risk

**DOI:** 10.3390/ijerph20146417

**Published:** 2023-07-20

**Authors:** Marlien Pieters, Iolanthe M. Kruger, Herculina S. Kruger, Yolandi Breet, Sarah J. Moss, Andries van Oort, Petra Bester, Cristian Ricci

**Affiliations:** 1Centre of Excellence for Nutrition, Faculty of Health Sciences, North-West University, Potchefstroom 2520, South Africa; marlien.pieters@nwu.ac.za (M.P.); salome.kruger@nwu.ac.za (H.S.K.); 2SAMRC Extramural Unit for Hypertension and Cardiovascular Disease, Faculty of Health Sciences, North-West University, Potchefstroom 2520, South Africa; 3Africa Unit for Transdisciplinary Health Research, Faculty of Health Sciences, North-West University, Potchefstroom 2520, South Africa; lanthe.kruger@nwu.ac.za (I.M.K.); 21195706@nwu.ac.za (Y.B.); petra.bester@nwu.ac.za (P.B.); 4Centre of Excellence for Hypertension in Africa Research Team, Faculty of Health Sciences, North-West University, Potchefstroom 2520, South Africa; 5Physical Activity, Sport and Recreation Research Focus Area, Faculty of Health Sciences, North-West University, Potchefstroom 2520, South Africa; hanlie.moss@nwu.ac.za (S.J.M.); abie.vanoort@nwu.ac.za (A.v.O.)

**Keywords:** Cox model, population attributable risk, model selection

## Abstract

When the Cox model is applied, some recommendations about the choice of the time metric and the model’s structure are often disregarded along with the proportionality of risk assumption. Moreover, most of the published studies fail to frame the real impact of a risk factor in the target population. Our aim was to show how modelling strategies affected Cox model assumptions. Furthermore, we showed how the Cox modelling strategies affected the population attributable risk (PAR). Our work is based on data collected in the North-West Province, one of the two PURE study centres in South Africa. The Cox model was used to estimate the hazard ratio (HR) of mortality for all causes in relation to smoking, alcohol use, physical inactivity, and hypertension. Firstly, we used a Cox model with time to event as the underlying time variable. Secondly, we used a Cox model with age to event as the underlying time variable. Finally, the second model was implemented with age classes and sex as strata variables. Mutually adjusted models were also investigated. A statistical test to the multiplicative interaction term the exposures and the log transformed time to event metric was used to assess the proportionality of risk assumption. The model’s fitting was investigated by means of the Akaike Information Criteria (AIC). Models with age as the underlying time variable with age and sex as strata variables had enhanced validity of the risk proportionality assumption and better fitting. The PAR for a specific modifiable risk factor can be defined more accurately in mutually adjusted models allowing better public health decisions. This is not necessarily true when correlated modifiable risk factors are considered.

## 1. Introduction

The availability of epidemiological and clinical data is accumulating quickly due to current technological developments [1,2]. State-of-the-art clinical instrumentations based on the paradigm of the internet of things and the possibility of conducting online surveys and interviews are just some of the many ways that allow the real-time collection of a huge amount of data [3]. This massive quantity of information is then helpful in many ways to improve human health [3,4,5]. For example, an early application of biostatistics can use the information collected during an epidemiological observation to better understand the association between different determinants of health and a given outcome. This approach is very common, especially addressing some causal relationships using prospective studies. In this field, one of the most common paradigms used for data analysis is the so-called survival analysis, an approach aimed to modelling the occurrence of events over the observational time [6]. Among many paradigms of survival analyses, the Cox model is the most frequently used for numerous reasons. Among the others, the Cox model considers multiple confounders, a typical problem in observational settings [7]. Briefly, the Cox model is a regression paradigm that allows to investigate the association between the hazard function *λ*(*t*|*Xi*) and a given set of covariates *Xi*. The Cox model is represented by the exponential function:$$\lambda \left(t|Xi\right)=\lambda 0\left(t\right)exp\sum_{i=0}^{n}\beta iXi$$

Considering two subjects with covariates *Xj* and *Xk*, the ratio of the hazards is
λtXjλtXk=λ0texpXjβλ0texpXkβ=exp((Xj−Xk)β)

The right-hand-side of the equation is not dependent on time, as the only time-dependent factor, *λ*0(*t*), was cancelled out. Therefore, the Cox model has the assumption that the baseline hazards of two subjects are equal irrespective of the covariate(s), so that the hazard ratio of two subjects is a constant. This assumption is referred to as the proportionality of risk assumption. The evaluation of this assumption is a critical step when applying the Cox model. The Cox model also assumes linearity between the logarithm of the hazard function and the covariates. The validity of this assumption increases with the model’s fitting. Another advantage of the Cox model is that its coefficients are easily interpreted as hazard ratios, the ratio of outcome occurrence between two groups over the observational time.

Epidemiological research has equipped itself with another tool, the population attributable risk (PAR), defined as the percentage of events that would be prevented if a given risk factor was eliminated from the target population [8]. The PAR is computed using the risk of a given outcome that is associated with a given exposure, and by the prevalence of the exposure [8,9]. Notably, being able to estimate the percentage of avoidable events due to a specific exposure is a very useful tool to guide policymakers to formulate public health interventions. Numerous scientific works use the Cox model to investigate to what extent certain exposures are associated with a given outcome. However, some recommendations about the choice of the time metric and the model’s structure are often disregarded causing lack of the proportionality of risk assumption [7,10,11,12]. For the same reason, a study may fail to frame the real impact of a risk factor in the target population.

Our aim was to show how modelling strategies affected Cox model performance. To this end, we showed how different Cox modelling strategies met the assumption of hazards proportionality, affected the model’s fitting, and influenced the estimates of the hazard ratio. Furthermore, we showed how the modelling strategies determined the estimates of the PAR. In this study we used the data from the South African leg of the Prospective Urban and Rural Epidemiology (PURE) study merged with the South African mortality registry. The Cox model analysis was applied to all-cause mortality in relation to smoking, alcohol use, low-physical activity level, and hypertension being representative of acknowledged modifiable risk factors.

## 2. Methods

### 2.1. The South African Leg of the PURE Study

The Prospective Urban and Rural Epidemiology (PURE) study is a research project aimed to investigate the association between risk factors and incidence of chronic disease in 27 low-, middle-, and high-income countries. Our paper is based on data collected in the North-West Province, one of the two PURE study centres in South Africa.

The target population is Setswana speaking, Black men, and women older than 30 years. Self-reported prior cardiovascular events, acute illness, pregnancy, or lactation were the exclusion criteria. The recruitment was conducted by a stratified random sample from 6000 randomly selected households located in rural and urban communities of the North West Province. The urban stratum was defined by established townships near a major city while the rural stratum was defined by communities under tribal law and more than 50 km from any major urban centre.

### 2.2. Data Collection

All self-reported information was collected by structured interview. Data regarding use of medication, alcohol, and tobacco were collected using a customized questionnaire, while physical activity was determined using the adapted BAECKE questionnaire [13]. Categories were created for ever use of alcohol and tobacco; low-physical activity level was defined for individuals with an overall BAECKE score below the first quartile. Brachial blood pressures were measured in the supine position at rest, by the Omron HEM-757 device (Omron Healthcare, Kyoto, Japan). Hypertension was defined according to the 2018 ESC/ESH guidelines [14] as a systolic or diastolic blood pressure equal to or greater than 140 mmHg or 90 mmHg, respectively, or by the use of anti-hypertensive medication. The study complies with the revised Helsinki Declaration and obtained full approval by the Health Research Ethics Committee of the North-West University (NWU), South Africa (04M10 and NWU-00016-10-A1). All participants signed an informed consent for the data treatment.

### 2.3. Statistical Methods

The study sample was described using the median and 5th to 95th range for continuous variables, counts, and percentages for categories. The description was provided for the overall analytical sample and by outcome status as recorded at the end of the follow-up. The comparison between survivors and the deceased was performed using a Mann–Whitney U-test and the ordinary *χ*^2^ test, for continuous and categorical data, respectively.

The Cox model was the paradigm used to estimate the hazard ratio (HR) of the outcomes associated with the exposure. Different types of models were used to estimate the risk of all-cause mortality in relation to smoking, alcohol use, physical inactivity, and hypertension. Firstly, a series of univariate models was considered. In this first evaluation, we used a Cox model with time to event as the underlying time variable. Secondly, we used a Cox model with age to event as the underlying time variable [10,11,12].

Briefly, the time to event is the ordinary metric of time used in time to event analysis; it is calculated as time difference (more commonly in days) between the event or the censor and the starting of observation. Age to event is still a time metric but it is coded as the age at which the event or the censor occurred. Scientific evidence supports the use of age to event as the time metric, especially for epidemiological research [10,11,12].

Finally, the above-mentioned model with age to event as the underlying time variable was implemented with age classes and sex as strata variables defining the baseline risk of the outcomes. For this last model, five age groups were considered as strata, a first group below 45 years, and four other 10-year groups with the last one constituted by participants older than 75 years of age. Notably, these age classes seem recommendable when creating age strata in the Cox model according to the mortality rates in South African adults where adult mortality seems quite constant below age 45, then it increases by 10-year classes [15,16].

Afterwards, we used a multivariate, mutually adjusted model considering all the risk factors described above. Here, we also considered time and age to event as underlying time variables, along with age classes and sex as strata variables. The pairwise correlation coefficient between covariates was reported in Appendix A.

To investigate the model’s proportionality of hazard assumptions, we performed a statistical test according to the multiplicative interaction term of all exposures present in the model and the log transformed time to event metric. The model’s fitting was investigated by means of the Akaike Information Criteria (AIC), an index interpretable as an estimator of prediction error defined as
AIC = 2k − 2ln (L)
where k is the number of model’s parameter and L is the maximized value of the likelihood function of the model, meaning the joint probability of the observed data in relation to the parameters of the model [17,18].

The AIC was chosen over other fitting criteria because it allows comparison of different models irrespective of the number of model parameters or structure [18]. Notably, the AIC is widely used in epidemiological research irrespective of limitation and valid alternatives, especially for the estimates of hazard functions [19,20,21,22,23].

The population attributable risk was performed using the sample prevalence, the risk estimates, and the covariance matrix of the coefficients in the case of the mutually adjusted model. Here, the approach indicated by Spiegelman et al. was chosen to compute the PAR and its 95% confidence limits [15]. For multivariate models, the partial PAR for a given exposure was performed, keeping the other variables as fixed so that it corresponds to the PAR attributable to that specific variable only.

All the statistical analyses were performed using the PHREG procedure of the SAS software verision 9.4. The %par macro of the SAS software was the tool used to compute the population attributable risk and its 95% confidence limits [24]. The example code for the analyses is reported in Appendix A; the %par macro is reported in Appendix A.

## 3. Results

This study is based on 1921 participants with full information about age, sex, smoking status, alcohol use, physical activity, and hypertension. The analytical sample was obtained after the exclusion of 88 subjects with missing data (4.4%) starting from a total sample size of 2009 participants.

The median observational time was 13 years (5th to 95th range = 2.5 to 13.6 years) corresponding to 21,532 person-years. The median age was 48 years (5th to 95th range = 36 to 69 years). There were 719 (37.4%) men and 951 participants were from a rural area (49.5%). More than half of the participants were current or former smokers (N = 1261, 65.8%) and 984 (51.4%) were declared to be current or former alcohol consumers. The physical activity score had a median value of 7.3 (5th to 95th range = 4.6 to 10.1). About half of the participants had hypertension (N = 907, 47.2%).

At the end of the observational time, there were 577 deaths (all-cause). All the elements under analysis had a different distribution between the survivors and deceased, resulting in statistically significant univariate comparisons. Baseline descriptions for the overall study sample and univariate comparisons between the survivors and deceased are reported in Table 1.

### Analysis of Death Determinants and Model Comparisons

When looking at the validity of risk proportionality assumption, we observed that univariate models with age as the underlying time variable performed much better than univariate models considering time to event. Moreover, adding age and sex as strata variables increases the validity of the risk proportionality assumption. Specifically, when considering smoking, alcohol use, and hypertension, we observed significant and borderline non-significant effects related to all the exposures. Here, when age was the underlying time variable, we no longer observed a statistically significant effect of the multiplicative term of the element under analysis with the logarithm of the time for smoking (*p* = 0.92) and alcohol (*p* = 0.86). On the contrary, the validity of risk proportionality assumption is still questionable for hypertension (*p* = 0.003). However, the risk proportionality assumption is valid also for hypertension (*p* = 0.14) when age classes and sex are added to the model as the strata. Furthermore, we reported that using age as time to event analysis greatly improves the model fitting with respect to time to event models. This evidence is even more noticeable when looking at models with age classes and sex as strata factors.

Notably, we observed an inconsistent pattern of the HR estimates when shifting from the simpler model with time to event to the model with age as the underlying time variable. We observed a 10% increased risk for smoking (from 1.34 (1.12, 1.61) to 1.44 (1.20, 1.72)) and for alcohol use (from 1.64 (1.38, 1.94) to 1.77 (1.49, 2.09)) when considering the simpler model with time to event with respect to the model with age as the underlying time variable. This pattern was not confirmed for low-physical activity and hypertension where the HR reduces, not being significant from 1.88 (1.58, 2.23) to 1.11 (0.93, 1.32) for low-physical activity and from 1.47 (1.25, 1.74) to 0.94 (0.79, 1.10) for hypertension, in the Cox model with time as the underlying time variable and age as the underlying time variable, respectively.

Similarly, we did not observe a relevant reduction in the HR estimates when comparing the models with age classes and sex strata to those without. In the univariate model for smoking, we reported a HR of 1.27 (1.05, 1.52), while the univariate model for alcohol use gave an HR of 1.44 (1.20, 1.72). Notably, having age classes and sex as strata factors reestablished the statistical significance of the HR for physical inactivity (HR = 1.41 (1.17, 1.68)) and hypertension (HR = 1.22 (1.03, 1.44)).

Finally, the higher HRs corresponded to the higher PAR. In conclusion, models with age as underlying time variables and age and sex as strata factors were the most adequate, because they better conformed to the proportionality of hazards’ assumption and had better model fitting.

A similar result was obtained for multivariate adjusted models. Briefly, multivariate-adjusted time to event models seem to have the worst performance by proportionality of hazards assumption when compared to models with age as the underlying time metric. This is true for smoking (*p* < 0.0001), alcohol use (*p* = 0.020), and hypertension (*p* = 0.036) with physical inactivity being questionable, still having a borderline non-significant *p*-value for the interaction term of sedentariness by the logarithm of time (*p* = 0.071). The proportionality of hazards assumption is fully achieved when using age as the underlying time variable, irrespective of having age classes and sex as strata variables. We report that mutually adjusted models had a better fitting compared to their univariate counterparts. However, this improvement seems to be modest only.

We observed a generalized reduction of HR estimates in multivariable adjusted models in relation to the adjusting effects of supplementary covariates. The correlation between covariates seems modest, ranging from 0.017 to 0.136 for all covariate pairs apart from smoking and alcohol use having a pairwise correlation of 0.439 (0.402, 0.474). Likely as a consequence, the statistically significant association between smoking and mortality was lost in all the mutually adjusted models. For smoking, the HR of the mutually adjusted model with age as the underlying time variable and age classes and sex as strata factors was 1.13 (0.92, 1.39). On the contrary, for alcohol use, the same HR was confirmed as statistically significant (HR = 1.28 (1.05, 1.58)). In mutually adjusted models with age as the underlying time variable and age classes and sex as strata, the physical inactivity was confirmed to be statistically related to mortality (HR = 1.34 (1.11, 1.60)). In the same type of models, hypertension was not confirmed as statistically significant (HR = 1.15 (0.97, 1.36)).

Mutual adjustment led to a generalized reduction in PAR because of the HRs’ reduction. Non-significant PARs were reported in correspondence to non-statistically significant HRs. Model characteristics are reported on Table 2.

## 4. Discussion

In the present work, we showed how to conduct an analysis using the Cox model to estimate hazard ratios and how to perform and interpret the resulting PAR. Certain technical aspects of this type of analysis emerged and should be considered by researchers before applying the Cox model. Firstly, we showed that the time to event is a questionable time metric for this analysis. This is particularly true in epidemiological research, or other real-setting situations, where the age of the participants is heterogeneous and is correlated to the occurrence of the outcome [10,11,12]. This is certainly true for mortality, as is evident by the univariate analysis applied to our data that showed that the median age of the survivors was statistically lower than those who passed away. This is also true for most of the commonly investigated non-communicable disease outcomes such as cardiovascular disease, cancer, and other morbid statuses commonly observed in the elderly. Moreover, it was reported that numerous technical features, such as statistical power and the risk of other biases, can be improved when a specific time variable is considered as the underlying time metric [10,11,12]. An important finding was that adjusting for an alternative time metric does not necessarily result in better model performance; sometimes the performance of models can be even worse [10,11,12]. Another important feature of the Cox model is the strata factors. In the Cox model the hazard function is composed of two parts. A baseline hazard, a locked element that is specific for a certain group of individuals, and the covariates, the variable part. Briefly, the baseline hazard represents the underlying risk of the event over time, without accounting for any covariate effects. It is then intuitive to expect that including age and sex would improve the model’s performance as they likely relate to different baseline hazards [6]. The recommendation is that any scientist applying the Cox model should consider using appropriate time to event variable and strata factors. It is also recommended to apply different Cox models with different time to event variables and strata factors and then choose the best model according to the validity of the model assumptions. Notably, in our work we observed how enhancing the complexity of the model by means of choice of time to event metric and baseline risk factors results in a reduction in the HR and the resulting PAR. This evidence pointed out that the most complex model is also the more conservative in terms of the null hypothesis. Notably, the most conservative approach should always be chosen in human research because this reduces the rate of false-positive results [25,26]. When looking at the analysis conducted on multivariate adjusted models, the above reported evidence was largely confirmed. Moreover, it is important to notice that the use of mutually adjusted models reduces the number of statistically significant HR estimates. Specifically, in our mutually adjusted models, we reported that the statistical significance was lost for smoking (irrespective of the underlying time metric) and for hypertension (when age was the underlying time variable). This is due to the adjusting effect, meaning that only the independent contributors to mortality risk remain statistically significant when many risk factors are considered at the same time. This resulted in a non-significant statistical effect for related PARs. Moreover, the PAR for a risk factor reduces its magnitude when the other risk factors are fixed, similarly to what happens in multivariate adjusted models. Adjusting for confounders in regression allows for the identification of independent contributors to the outcomes. No less, here we showed that multivariate adjusted models may allow us to obtain more accurate PARs. A multivariate adjusted PAR estimate, however, could lead to wrong public health decisions. For example, let us imagine that a health promotion programme is conducted on this same target population. This analysis showed that it is better to focus resources on public health programmes that reduce alcohol use and physical inactivity. It must be noticed that such decisions should be taken with responsibility as a choice only based on statistical significance lost due to collinearity can limit the effect of the public health intervention. On the one hand, it seems more efficient to aim the public health intervention towards those modifiable risk factors that were statistically significant. On the other hand, it is more cautious, and indeed recommendable, to act also on likely relevant modifiable risk factors, even irrespective of their statistical significance. Referring to the above example from our data, alcohol is retained towards smoking because it is more strongly related to the outcomes and statistically significant in the mutually adjusted model. Moreover, alcohol and smoking are strongly correlated behaviors. On the one hand, acting on alcohol use would indeed reduce the risk of the outcome. On the other hand, a public health intervention acting only on alcohol use would not have any effect on those who smoke but do not use any alcohol. Alcohol and tobacco use are generally strongly correlated, this collinearity may lead to the exclusion of one of the two variables. In this perspective, the PAR calculation is still rigorous; however, it may represent a sneaky suggestion for public health policy makers.

### 4.1. Strengths and Limitations

This work has numerous strengths. Firstly, it provides certain useful suggestions regarding how to perform a Cox model and how to estimate the PAR, a measure of great epidemiological usefulness. Secondly, in our work we not only provide recommendations, but we also demonstrate the validity of our arguments based on real data. Finally, the quality of our data collection strategy indeed represents a strength of our work. Some limitations of our work should also be highlighted. Firstly, we do not present any results based on simulated data. On the one hand, the numerical simulation represents a rigorous methodology for it provides data containing specific evidence defined by the researchers. On the other hand, effective simulation of a real-life phenomenon, such as the correlation among factors and bias of various types, is somehow questionable. Moreover, our results agree with results from simulated data reported in previous publications [11,12]. Another limitation of our study could be the lack of result generalizability due to the specificity of our target population. We did not intend to provide generalizable results about these specific modifiable risk factors. The epidemiological evidence reported here is clearly not extendable to other target populations as the South African population is quite unique. However, we do believe that the methodological evidence provided here may represent a valid input to better address Cox model analysis and its application to perform the PAR.

### 4.2. Conclusions

The choice of the time to event metric is critical in Cox models. Here we showed that age to event is better than time to event to guarantee the assumption of the proportionality of risks and the model’s fitting. Another critical factor is the choice of the factors that determine the baseline hazard. In our work, we showed that using age classes and sex also improves the assumption of the proportionality of risks and the model’s fitting.

The Population Attributable Risk for a specific modifiable risk factor can be defined in mutually adjusted models to consider the correlation between the different risk factors that determine the hazard of the outcome. This approach allows for the estimation of the PAR by specific risk factors, allowing better public health decisions. However, this is not necessarily true when correlated modifiable risk factors are considered.

## Figures and Tables

**Table 1 ijerph-20-06417-t001:** Baseline characteristics of the total sample and by mortality status.

	All (N = 1921)	Survivors (N = 1344)	Deceased (N = 577)	* *p*-Value
Person-year	21,532	17,711	3821	Not Applied
Median Fu (years)	13.0 (2.5, 13.6)	13.2 (12.7, 13.6)	6.6 (0.7, 12.4)	Not Applied
Age (years)	48.0 (36.0, 69.0)	47.0 (36.0, 67.0)	52.0 (37.0, 74.0)	<0.0001
<45 years (N, %)	700 (36.4)	558 (41.5)	142 (24.6)	<0.0001
45 to 54 years (N, %)	643 (33.5)	449 (33.4)	194 (33.6)	0.9321
55 to 64 years (N, %)	389 (20.3)	245 (18.2)	144 (25.0)	0.0007
65 to 74 years (N, %)	148 (7.7)	77 (5.7)	71 (12.3)	<0.0001
≥75 years (N, %)	41 (2.1)	15 (1.1)	26 (4.5)	<0.0001
Men (N, %)	719 (37.4)	436 (32.4)	283 (49.1)	<0.0001
Smokers (N, %)	1261 (65.8)	849 (63.3)	412 (71.5)	0.0005
Alcohol users (N, %)	984 (51.4)	630 (47.0)	354 (61.7)	<0.0001
Physical activity score	7.3 (4.6, 10.1)	7.6 (4.7, 10.2)	6.4 (4.3, 9.7)	<0.0001
Physically inactive (N, %)	481 (25.0)	275 (20.5)	206 (35.7)	<0.0001
Hypertension (N, %)	907 (47.2)	589 (43.8)	318 (55.1)	<0.0001

Notes: *: *p*-value for comparison of survivors vs. deceased. Test performed the using Mann–Whitney U test or the *χ*^2^ test for continuous or categorical data, respectively. Fu—follow-up.

**Table 2 ijerph-20-06417-t002:** Hazard ratio and PAR% for different univariate Cox models.

Model	Exposure	HR (95% CI)	PAR% (95% CI)	* *p*-Value	^‡^ AIC
Univariate model_1_		1.34 (1.12, 1.61)	18.3% (7.5, 28.6)	0.0002	8490.7
Univariate model_2_		1.44 (1.20, 1.72)	22.3% (11.7, 32.3)	0.9247	7563.2
Univariate model_3_	Smoking	1.27 (1.05, 1.52)	14.9% (3.6, 25.8)	0.8118	5666.0
Multivariate model_1_		1.12 (0.91, 1.37)	7.1% (−5.8, 19.8)	<0.0001	8378.9
Multivariate model_2_		1.13 (0.92, 1.39)	8.0% (−4.8, 20.6)	0.8251	7495.6
Multivariate model_3_		1.13 (0.92, 1.39)	7.8% (−5.2, 20.5)	0.1222	5619.3
Univariate model_1_		1.64 (1.38, 1.94)	24.0% (16.0, 31.8)	0.0815	8436.5
Univariate model_2_		1.77 (1.49, 2.09)	28.0% (20.0, 35.7)	0.8647	7507.3
Univariate model_3_	Alcohol use	1.44 (1.20, 1.72)	18.1% (9.1, 26.8)	0.4436	5637.7
Multivariate model_1_		1.28 (1.04, 1.57)	12.4% (2.2, 22.4)	0.0204	8378.9
Multivariate model_2_		1.28 (1.04, 1.57)	12.5% (2.2, 22.4)	0.5916	7495.6
Multivariate model_3_		1.28 (1.05, 1.58)	12.6% (2.3, 22.6)	0.4785	5619.3
Univariate model_1_		1.88 (1.58, 2.23)	16.8% (11.6, 21.9)	0.9373	8470.8
Univariate model_2_		1.11 (0.93, 1.32)	2.7% (−2.1, 7.5)	0.3788	7595.6
Univariate model_3_	Sedentariness	1.41 (1.17, 1.68)	9.7% (4.2, 15.1)	0.3818	5673.6
Multivariate model_1_		1.42 (1.18, 1.70)	10.0% (4.4, 15.4)	0.0709	8378.9
Multivariate model_2_		1.44 (1.20, 1.72)	10.4% (4.8, 15.9)	0.9466	7495.6
Multivariate model_3_		1.34 (1.11, 1.60)	8.2% (2.7, 13.5)	0.3915	5619.3
Univariate model_1_		1.47 (1.25, 1.74)	17.8% (10.2, 25.3)	0.0723	8499.4
Univariate model_2_		0.94 (0.79, 1.10)	Not-estimable	0.0026	7596.2
Univariate model_3_	Hypertension	1.22 (1.03, 1.44)	9.7% (1.3, 17.9)	0.1436	5681.8
Multivariate model_1_		1.20 (1.01, 1.42)	8.8% (0.3, 17.1)	0.0359	8378.9
Multivariate model_2_		1.19 (1.00, 1.42)	8.7% (0.2, 17.0)	0.1003	7495.6
Multivariate model_3_		1.15 (0.97, 1.36)	6.8% (−1.7, 15.2)	0.1222	5619.3

Notes: * *p*-value for factor*log(time) *p*-value < 0.05 represents non-valid assumption for risk proportionality. ‡ Akaike information criterion (lower values correspond to better model fitting). Model_1_: Cox model with time to event as underlying time metric, Model_2_: Cox model with age to event as underlying time metric, Model_3_: Cox model with age to event as underlying time metric and age classes and sex as strata variables. Multivariate models are mutually adjusted.

## Data Availability

Data are available by request to the corresponding author.

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
