# Peer review of "Strategies of Modelling Incident Outcomes Using Cox Regression to Estimate the Population Attributable Risk"

_ijerph, 2023, doi:10.3390/ijerph20146417_

Round 1
Reviewer 1 Report
The research article is of applied nature and has used a dataset regarding the mortality and some of its possible risk factors to fit and identify the so called “best” model. The article is good in the sense that it uses a real data for analysis that can lead to some decision making and making some policy in the future for a specific geographical location.
However, following issues should be addressed before the recommendation of this article for publication.
Major:
1. The whole article focuses on a real data which is actually missing. The complete data must be made public along with its missing observations (as a supplementary material) as mentioned in P4, L150 of the text. Only in this case, the genuineness and true picture of the analysis and results can be established.
2. It is nowhere mentioned that what was actual sample size out of which 1921 was considered in the paper after ignoring missing data.
3. The predictors/risk factors are not clearly defined. It should be mentioned that which covariates are categorical each with its number and names of categories and which are quantitative. The justification and rationale of converting a continuous variable (e.g., age here) into categorical variable is also required.
4. Why median is used as an average instead of mean for continuous data? Also, “5th to 95th range for continuous variables” on P3, L112 requires explanation. Also, the dispersion in the quantitative variables should also be reported along with averages. Similarly, title of Table 1 should be self-explanatory. It is better to report continuous and categorical variables separately even in the same table.
5. The p-value in the last column of Table 1 corresponds to which hypothesis and why? Why do you want to compare survivors and deceased regarding different variables and even each of different categories separately for a particular risk factor. What kind of information, the authors want to gain and how it will be helpful in fitting Cox regression.
6. The pairwise correlation of different risk factors should be calculated, depending on the nature of the variables, and reported in the text.
7. When there are more than one risk factors, fitting model with one risk factor at a time is not logical and can provide misleading results also. Because the correlation between all may change the results when all of them are included in the model. One reflection of such scenario can be seen in Table 2, where Alcohol use is significant in multivariate case but insignificant in univariate case. So “so called” univariate case will provide misleading results in this case.
8. In Table 2, why PAR% C.I. is not estimable for hypertension in univariate Model 2? It should be discussed and explained.
9. Why the focus is on “so called” univariate and multivariate case? What will be the situation if a model with combination of any two or three risk factors is built? Also, what about the significance of interaction terms?
10. The results of testing the proportional hazard assumption are not given for any model.
11. Goodness of fit of any model in comparison to the full model is not tested. The Likelihood ratio test and/or coefficient of concordance and its Confidence Intervals can be used.
12. Hazard ratios should be discussed and explained in the text in detail.
13. If Kaplan Meier Survival Curve is used along with showing of 95% CI coverage, it will give a clearer picture. Also, the number of censored and un-censored observations should be mentioned with the HR results.
14. Although, there are already only four risk factors in the study, but what will be the situation if some variable selection method or significance of some risk factors is implied in fitting of full model with all risk factors.
Minor:
1. The language should be improved. There are many sentences that need to be rephrased for a clear understanding.
2. The CI limits within parentheses should be separated by comma instead of semicolon.
3. There should be a table which should reflect name of each fitted model, name of response variable, and name(s) of independent variables/risk factors in that model.
4. The latest references are missing in the article. The latest reference that authors have used in the manuscript is from 2020. There is a huge literature related to survival models available even for year 2023.
The language should be improved. There are many sentences that need to be rephrased for a clear understanding.
Author Response
Major:
- The whole article focuses on a real data which is actually missing. The complete data must be made public along with its missing observations (as a supplementary material) as mentioned in P4, L150 of the text. Only in this case, the genuineness and true picture of the analysis and results can be established.
R: We will discuss with the journal if the data have to be shared and in which form. However, considering the original sample size (2,009) missing represents below 5%. On the one hand below 5% of missing does not generally represent a problem for data analysis. On the other hand, the data analysis is here a tool to show the performances of different types of Cox models and should not have any epidemiological interest. Moreover, the association between alcohol use, smoking, hypertension, and low physical activity with all-cause mortality are nothing new in the epidemiological field as specified in the draft (lines 255-261 and 340-341)
- It is nowhere mentioned that what was actual sample size out of which 1921 was considered in the paper after ignoring missing data.
R: lines 180-183 specifically state it, on the same line we also added the starting sample size of 2,009
- The predictors/risk factors are not clearly defined. It should be mentioned that which covariates are categorical each with its number and names of categories and which are quantitative. The justification and rationale of converting a continuous variable (e.g., age here) into categorical variable is also required.
R: details have been added about the age (Lines 145-147). The first version of the manuscript details quite well the definition of low physical activity and hypertension (Lines 114-115). Details about the categories of alcohol and tobacco use as been added (Lines 114-117)
- Why median is used as an average instead of mean for continuous data? Also, “5thto 95th range for continuous variables” on P3, L112 requires explanation. Also, the dispersion in the quantitative variables should also be reported along with averages. Similarly, title of Table 1 should be self-explanatory. It is better to report continuous and categorical variables separately even in the same table.
R: The median is a generalized index of position that is not affected by variable skewness so we adopted it for this study. The 5th to 95th range is a useful dispersion index. It excludes 10% of extreme observations and it is wider that an interquartile range. I used it in previously published papers without any particular question from reviewers https://www.ncbi.nlm.nih.gov/pmc/articles/PMC5972779/
- The p-value in the last column of Table 1 corresponds to which hypothesis and why? Why do you want to compare survivors and deceased regarding different variables and even each of different categories separately for a particular risk factor. What kind of information, the authors want to gain and how it will be helpful in fitting Cox regression.
R: the test (U-test or χ2) between survivors and deceased represents a univariate raw comparison the null hypothesis of these tests is basic knowledge for the readers. It is interesting for the reader to observe a difference between the two groups. Also, it is interesting for the reader to notice if such significance is maintained in the Cox model.
- The pairwise correlation of different risk factors should be calculated, depending on the nature of the variables, and reported in the text.
A supplementary table of correlation has been added, results of the correlations has been added lines (lines 151-152 and 251-253).
- When there are more than one risk factors, fitting model with one risk factor at a time is not logical and can provide misleading results also. Because the correlation between all may change the results when all of them are included in the model. One reflection of such scenario can be seen in Table 2, where Alcohol use is significant in multivariate case but insignificant in univariate case. So “so called” univariate case will provide misleading results in this case.
R: It is our aim to show to the reader the misleading results from univariate models. The entire work does not aim to show results, but it aims to show how to conduct the modelling, also showing the consequences of wrong approaches.
- In Table 2, why PAR% C.I. is not estimable for hypertension in univariate Model 2? It should be discussed and explained.
R: the Relative risk for hypertension is below 1, in this case the PAR cannot be calculated (if the risk is protective the population attributable risk represents an increase of mortality for the elimination of the risk factor which has no epidemiological sense). We used “not estimable” as it is the output of the %par in such situations
- Why the focus is on “so called” univariate and multivariate case? What will be the situation if a model with combination of any two or three risk factors is built? Also, what about the significance of interaction terms?
R: The multivariate model is mutually adjusted, and this represents the next level of complexity from univariate model. Investigation of interaction is not of our interest. Again, this work does not aim to show results but adequacy of model assumptions in different cases. To our knowledge the inclusion of interactions does not affect model fitting or the assumption of risk proportionality.
- The results of testing the proportional hazard assumption are not given for any model.
R: The Pvalue of table 2 represents exactly this. The table’s footnote is “*P-value for factor*log(time) P-value < 0.05 represents non-valid assumption for risk proportionality.”
- Goodness of fit of any model in comparison to the full model is not tested. The Likelihood ratio test and/or coefficient of concordance and its Confidence Intervals can be used.
R: We report the AIC as a measure of fit. The alternative proposed by the reviewer (likelihood ratio test) is in general a valid alternative and it is preferable to AIC for nested models. Notably, our models are not nested so the AIC should be preferred. The null model (model with only the intercept) in the case of the Cox model is not of great interest as the intercept is the baseline risk.
- Hazard ratios should be discussed and explained in the text in detail.
R: We added a brief description of HRs in relation to the different models (lines 214-221 and 229-232).
- If Kaplan Meier Survival Curve is used along with showing of 95% CI coverage, it will give a clearer picture. Also, the number of censored and un-censored observations should be mentioned with the HR results.
R: the KM survivor curves are not adequate for describe results from Cox analyses. Firstly, KM does not consider the adjusting factors and the strata, so it is also not useful for our models. Secondly, KM curves can be used to evaluate the proportionality of risks assumption. However, it is a visual inspection and so it is qualitative and questionable. The test proposed in our manuscript is more appropriate.
- Although, there are already only four risk factors in the study, but what will be the situation if some variable selection method or significance of some risk factors is implied in fitting of full model with all risk factors.
R: The aim of our study is only to show the Cox assumptions in relation to the choice of time metric and strata factor. In this perspective, increasing the number of covariates does not affect our results. The use of variable selection approaches may influence Cox’s assumptions of linearity but this is far beyond the scope of our draft.
Minor:
- The language should be improved. There are many sentences that need to be rephrased for a clear understanding.
R: The manuscript has been revised
- The CI limits within parentheses should be separated by comma instead of semicolon.
R: Semicolon has been substituted by a comma
- There should be a table which should reflect name of each fitted model, name of response variable, and name(s) of independent variables/risk factors in that model.
R: We respectively disagree, the coding used to describe the models is efficient and easy to understand.
- The latest references are missing in the article. The latest reference that authors have used in the manuscript is from 2020. There is a huge literature related to survival models available even for year 2023.
R: Some more recent references has been added
Among the others: Choice of Time-Scale in Time-to-Event Analysis: Evaluating Age-Dependent Associations. Ann Epidemiol 2021

Reviewer 2 Report
The paper is interesting and it can be published after the following comments:-
1- The authors should increase the literature of review.
2- The formulas of AIC should be added.
3- Future works should be added.
4- Some papers used AIC and P-value measures should be cited as
. - Bantan, R.A.R.; Chesneau,C.; Jamal, F.; Elbatal, I.; Elgarhy, M. The Truncated Burr X-G Family of Distributions: Properties and Applications to Actuarial and Financial Data. Entropy 2021, 23, 1088.
- Bantan, R.A.R.; Jamal, F.;Chesneau, C.; Elgarhy, M. Theory and Applications of the Unit Gamma/Gompertz Distribution. Mathematics 2021, 9, 1850
- R. A. ZeinEldin, C. Chesneau, F. Jamal, and M. Elgarhy,A. M. Almarashi and
S. Al-Marzouki, Generalised truncated Fr ́echet generated family distributions
and their applications, ”Computer Modeling in Engineering
and Sciences, vol. 126,no. 2, pp. 791–819, 2021.
- Aldahlan, M.A.; Jamal, F.; Chesneau, C.; Elbatal, I.; Elgarhy, M. Exponentiated power generalized Weibull power series family of distributions: Properties, estimation and applications. PLoS ONE 2020, 15, e0230004. https://doi.org/10.1371/journal.pone.0230004
-Al-Marzouki, S.; Jamal, F.; Chesneau, C.; Elgarhy, M. Type II Topp–Leone power Lomax distribution with applications. Mathematics2020, 8, 4.
- Bantan, R.A.R.; Chesneau, C.; Jamal, F.; Elgarhy, M.; Tahir, M.H.; Ali, A.; Zubair, M.; Anam, S. Some New Facts about the Unit-Rayleigh Distribution with Applications. Mathematics 2020, 8, 1954
-
The paper is written well
Author Response
1- The authors should increase the literature of review.
R: The literature review was implemented also according to your valuable suggestions.
2- The formulas of AIC should be added.
R: the formula of the AIC was added (lines 155-163 )
3- Future works should be added.
R: Actually we do not foresee future works in this specific aspect
4- Some papers used AIC and P-value measures should be cited as
.- Bantan, R.A.R.; Chesneau,C.; Jamal, F.; Elbatal, I.; Elgarhy, M. The Truncated Burr X-G Family of Distributions: Properties and Applications to Actuarial and Financial Data. Entropy 2021, 23, 1088.
- Bantan, R.A.R.; Jamal, F.;Chesneau, C.; Elgarhy, M. Theory and Applications of the Unit Gamma/Gompertz Distribution. Mathematics 2021, 9, 1850
- R. A. ZeinEldin, C. Chesneau, F. Jamal, and M. Elgarhy,A. M. Almarashi and
S. Al-Marzouki, Generalised truncated Fr ́echet generated family distributions
and their applications, ”Computer Modeling in Engineering
and Sciences, vol. 126,no. 2, pp. 791–819, 2021.
- Aldahlan, M.A.; Jamal, F.; Chesneau, C.; Elbatal, I.; Elgarhy, M. Exponentiated power generalized Weibull power series family of distributions: Properties, estimation and applications. PLoS ONE 2020, 15, e0230004. https://doi.org/10.1371/journal.pone.0230004
-Al-Marzouki, S.; Jamal, F.; Chesneau, C.; Elgarhy, M. Type II Topp–Leone power Lomax distribution with applications. Mathematics2020, 8, 4.
- Bantan, R.A.R.; Chesneau, C.; Jamal, F.; Elgarhy, M.; Tahir, M.H.; Ali, A.; Zubair, M.; Anam, S. Some New Facts about the Unit-Rayleigh Distribution with Applications. Mathematics 2020, 8, 1954
R: The papers were included as references when the AIC was introduced in methods.
Reviewer 3 Report
M Pieters etc. explored modeling strategies that affected Cox model assumptions. They tested different Cox modeling strategies and compared the performance. They showed how to conduct an analysis using the Cox model to estimate hazard ratios and how to perform and interpret the resulting PAR.
Minor revision:
- What is the difference between time and age to event? Does time refer to observational time? What is the starting point of observation?
- In line 142, “The par% macro”, however in line 144, it showed as “the %par macro”. I suggest using “The %par macro” in all the paper, at least, it should be consistent.
Author Response
M Pieters etc. explored modeling strategies that affected Cox model assumptions. They tested different Cox modeling strategies and compared the performance. They showed how to conduct an analysis using the Cox model to estimate hazard ratios and how to perform and interpret the resulting PAR.
Minor revision:
- What is the difference between time and age to event? Does time refer to observational time? What is the starting point of observation?
R: supplementary explanation has been added (lines 136-140).
- In line 142, “The par% macro”, however in line 144, it showed as “the %par macro”. I suggest using “The %par macro” in all the paper, at least, it should be consistent.
R: “The %par macro” as been used consistently in the manuscript (line 175)
Round 2
Reviewer 1 Report
It is really strange that authors, instead of addressing and responding the queries, mostly insisted that what they have done should be considered as correct.
1- The data is still not provided on the basis of which your whole manuscript is based. Your claim is that your focus is on performance of different models of your own choice. Actually, there is no novelty with respect to methodology. The only thing that can have some novelty is the real data and its analysis in true spirit which is missing in your article. It was not a big task to consider all models with two or three predictors in case of when you have only four predictors.
Also, what actually you want to show the readers? It is a universal truth that if predictors are correlated then considering one predictor at each time will potentially have different results/inferences than the situation when all correlated predictors are included in the model. In response to point 7, your claim is that your focus is not to show results but to show how to conduct modelling. But, I believe you are misguiding the readers in this regard too.
When you have a very few predictors in your study, at least first order interactions can be considered especially when your focus is modeling strategies.
2- Your claim in response to point 4 is that you are using median instead of mean because of skewed data. Again, your insistence is that reviewer and readers just believe without any proof in terms of data that your data is skewed. The data must be there as evidence to support your stance.
3- In response to point 5, again you are confusing the reader. You used the wording "univariate raw comparison.....". I can't understand what you actually want to show in your manuscript. As a statistician, when you are trying to identify a so called "best possible" model that is close to the ever-unknown true model, you are focusing again and again wrong strategy of using individual predictors at a time.
4- In response to point 5, you provided a supplementary table. Since you have a mixture of continuous, binary and multi-category predictors, different types of correlation measures should be used for different pairs. You have not mentioned the type of correlation measures but significance of some measure which is also significant in more than half pairs.
5- In response to point 10, your evidence is Table 2 where assumption is not met in some models? what the analyst will do and recommend in such case?
Author Response
1- The data is still not provided on the basis of which your whole manuscript is based. Your claim is that your focus is on performance of different models of your own choice. Actually, there is no novelty with respect to methodology. The only thing that can have some novelty is the real data and its analysis in true spirit which is missing in your article. It was not a big task to consider all models with two or three predictors in case of when you have only four predictors.
Also, what actually you want to show the readers? It is a universal truth that if predictors are correlated then considering one predictor at each time will potentially have different results/inferences than the situation when all correlated predictors are included in the model. In response to point 7, your claim is that your focus is not to show results but to show how to conduct modelling. But, I believe you are misguiding the readers in this regard too.
When you have a very few predictors in your study, at least first order interactions can be considered especially when your focus is modeling strategies.
R: We are very sorry; the PURE data cannot be shared because of internal policy. The aim of the paper was methodological, we acknowledge that this review does not recognize the novelty and the scientific interest of our work. We disagree for believe that our work is a step-by-step approach showing how changing the structure of the Cox Model improve the assumptions on which it is based.
2- Your claim in response to point 4 is that you are using median instead of mean because of skewed data. Again, your insistence is that reviewer and readers just believe without any proof in terms of data that your data is skewed. The data must be there as evidence to support your stance.
R: This is the first time that a reviewer requests us to share the data for the reader can check the skewness of the data. Anyway, the median and mean are the same if the data are normally distributed. If data are skewed, then the mean is biased. So, we do not understand why we should use the arithmetic average instead of the median.
3- In response to point 5, again you are confusing the reader. You used the wording "univariate raw comparison...". I can't understand what you actually want to show in your manuscript. As a statistician, when you are trying to identify a so called "best possible" model that is close to the ever-unknown true model, you are focusing again and again wrong strategy of using individual predictors at a time.
R: We respectively disagree because the aim of the study is particularly clear. Again, we show to the reader the effect of using different Cox model structure and their effect on the assumptions, this is a pedagogical approach that generally works. We are confident that the readers will appreciate it.
4- In response to point 5, you provided a supplementary table. Since you have a mixture of continuous, binary and multi-category predictors, different types of correlation measures should be used for different pairs. You have not mentioned the type of correlation measures but significance of some measure which is also significant in more than half pairs.
R: We used Spearman correlation the title of the table has been specified accordingly. Please consider that the use of point-biserial correlation or Phi does not represent a particular advantage (numerical equivalence) for that table has an explorative purpose
5- In response to point 10, your evidence is Table 2 where assumption is not met in some models? what the analyst will do and recommend in such case?
R: The recommendation would be to change to the next level with a model with better performance. We believe that it should be logical for the reader to deduce this solution. In general, the recommendation is to have age as underlying time variable and age classes and sex as strata factor.